# Maintaining School Foodservice Operations in Ohio during COVID-19: “This [Was] Not the Time to Sit Back and Watch”

**DOI:** 10.3390/ijerph19105991

**Published:** 2022-05-14

**Authors:** Ashlea Braun, Joshua D. Hawley, Jennifer A. Garner

**Affiliations:** 1Medical Dietetics, School of Health and Rehabilitation Sciences, The Ohio State University College of Medicine, Columbus, OH 43210, USA; ashlea.braun@okstate.edu; 2Department of Nutritional Sciences, School of Education and Human Sciences, Oklahoma State University, Stillwater, OK 74078, USA; 3John Glenn College of Public Affairs, The Ohio State University, Columbus, OH 43210, USA; hawley.32@osu.edu

**Keywords:** COVID-19, SARS-CoV-2, food services, diet quality, food insecurity, school foodservice, civil defense, food supply, nutrition policy, hunger

## Abstract

The COVID-19-related lockdowns led to school closures across the United States, cutting off critical resources for nutritious food. Foodservice employees emerged as frontline workers; understanding their experiences is critical to generate innovations for program operations and viability. The purpose of this cross-sectional study was to characterize COVID-19-related foodservice adaptations for summer and school year meal provision. Public school district foodservice administrators across Ohio were surveyed in December 2020. Questions related to meal provision before, during, and after COVID-19-related school closures. Results indicate the majority of districts continued providing meals upon their closure in Spring 2020 (*n* = 182, 87.1%); fewer did so in Summer (*n* = 88, 42.1%) and Fall (*n* = 32, 15.3%). In Spring and Summer, most districts that offered meals functioned as ‘open sites’ (67.0% and 87.5%, respectively), not limiting food receipt to district-affiliated students. Most districts employed a pick-up system for food distribution (76–84% across seasons), though some used a combination of approaches or changed their approach within-season. Qualitatively, districts reported both “successes” (e.g., supporting students) and “challenges” (e.g., supply chain). Despite being ill-prepared, districts responded quickly and flexibly to demands of the pandemic. This analysis provides insight for future practice (e.g., establishing community partnerships) and policy (e.g., bolstering local food systems).

## 1. Introduction

In 2019, schools across the United States (U.S.) served nearly five billion lunches and over two billion breakfasts to students during the school year and summer months [1]. Accordingly, school meals provide an estimated 47% of a child’s daily energy intake, while consumption of these meals is associated with improved diet quality, markers of cardiometabolic health, and academic performance [2,3]. Recent literature further highlights additional long-term benefits on eating behavior as well as influences on future food preferences and habits [3,4].

School meals are instrumental for the more than six million children experiencing food insecurity in the U.S., and for those children living in food insecure and marginally secure households [5,6]. Data indicate that for these high-risk children, school meals are associated with improved attendance, academic performance, and diet quality [7,8,9,10]. Given the critical importance of these school meals, landmark federal legislation (e.g., National School Lunch Program) has enabled delivery of an overwhelming number of meals free or at a reduced price (nearly 75% and 85% of breakfasts and lunches in 2019, respectively) [1]. 

Historical controversies surround school meals, including questions regarding the cost, quantity, and quality of options available, excessive food waste, and schools’ approaches to adhering to nutrition standards [11,12,13]. However, additional federal regulations have mandated meal pattern requirements and other policies to ensure school meals align with evidence-based dietary guidelines. Despite these controversies over time, data indicate that without this safety net, children face potentially detrimental deficits in provision of essential nutrition for growth and development [14,15,16,17].

Although school foodservice programs have remained a steady resource for school-aged children since their inception in the 1940s, a small body of literature has highlighted insufficient emergency preparedness among schools and potential negative ramifications of emergency scenarios on participating families [18,19]. This issue was brought to the forefront in March of 2020, when the condition caused by SARS-CoV-2, coronavirus disease 2019 (COVID-19), was declared a pandemic. This resulted in widespread public health responses, including nationwide lockdowns to limit spread of and exposure to the virus [20]. School closures followed, as well as simultaneous transition to remote modes of education delivery, leaving millions of children at home. 

Existing data regarding decreases in child nutrition, health, and well-being during the summer months allude to potential deleterious effects of this unprecedented period, including unstructured days and increases in obesity-related risk factors (e.g., increased consumption of added sugars and decreased consumption of vegetables) [21,22,23]. Indeed, preliminary data from the shutdowns point to increases in child body mass index (BMI) and BMI-for-age z-scores (zBMI), effects which may persist, or contribute to related comorbidities, into adulthood [24,25,26]. These effects may be even more profound in high-risk children living in food insecure households or in low-income communities with poor access to healthy food. 

At the onset of the pandemic, foodservice staff emerged as essential workers, continuing to serve food to children despite concerns over safety [27]. Accordingly, the United States Department of Agriculture responded to the COVID-19 shutdowns with legislation enabling relaxation of existing requirements to facilitate delivery of meals, extenuating circumstances notwithstanding. These included Meal Times, Non-congregate Feeding, and Meal Pattern Waivers enabling flexibility in the timing of when meals could be ‘served’, the provision of food in non-group settings, and the serving of food that did not adhere to meal pattern requirements, respectively. Initial findings reveal variability and creativity in school responses to these changes [28,29]. Various challenges faced by schools in navigating and implementing these modifications are characteristic of a “fragmented” and “fragile” system, resulting in widespread concerns surrounding schools’ ability to maintain safe foodservice operations, warranting innovation [15,29,30]. Given these difficulties, the public health community has called for dissemination of information regarding schools’ offering of meals during the pandemic-related closures [14]. Understanding school responses to and experiences with the COVID-19-related changes to foodservice is critical to inform future school foodservice policy and emergency preparedness. Accordingly, the purpose of this study was to characterize COVID-19-related foodservice adaptations, including impacts on both summer and school year meal provision, among public schools across the state of Ohio. The findings of this work are essential to informing innovations for equipping school foodservice programs to withstand future adversities and emergencies.

## 2. Materials and Methods

### 2.1. Study Design and Survey Development

For this cross-sectional study, faculty from The Ohio State University consulted with staff from the Ohio Department of Education (ODE) on the design of a comprehensive survey to capture the experiences of foodservice staff within Ohio school districts during the COVID-19-related school closures. The instrument was refined iteratively based on pilot testing and cognitive interviewing with key stakeholders from within the target population to achieve face validity. The final instrument included 33 items, with both closed-ended (i.e., quantitative) and open-ended (i.e., qualitative) questions for generation of rich data to capture both procedures and perceptions. Additional quantitative data regarding the number and types of meals served in 2020 were obtained from the Ohio Department of Education website (education.ohio.gov, accessed on 1 October 2021).

Given the dynamic nature of the closures and subsequent plausible service adaptations, questions were designed to assess the evolution of processes since before March of 2020. Question sets were focused on three seasons/periods of time: before (i.e., early Spring 2020), during (i.e., late Spring and Summer of 2020), and after (i.e., Fall of 2020) the COVID-19 related shutdowns (hereafter, Spring, Summer, and Fall, respectively). Specific questions were also designed to capture changes to foodservice operations responsive to federal COVID-19 legislation (e.g., Meal Pattern Waivers). The survey ended with four open-ended questions: (1) ‘Since March 2020, what factors (e.g., district assets, attributes, or challenges) have influenced your approach to COVID-related foodservice changes the most?’, (2) ‘As you reflect on your operational changes, what has gone well?’, (3) ‘What has been challenging?’, and (4) ‘What concerns do you have for the remainder of 2020–2021 school year?’ The final instrument and all related study procedures were approved by The Ohio State University Institutional Review Board (protocol #2020B0300).

### 2.2. Recruitment and Distribution 

The finalized survey was housed within and managed by the Center for Human Resource Research at The Ohio State University (CHRR). For distribution, an administrative representative from each Ohio public school district across all 88 counties in the state was emailed in early December 2020 and invited to complete the survey on CHRR’s online platform. Survey respondents were anonymous, but were asked to enter the name of the district on whose behalf they were responding. Recruitment emails included information regarding the study’s purpose, risks, and benefits; response to the email and embedded survey constituted informed consent. The survey remained open for three weeks and two reminders were sent to the entire contact list at one and two weeks post-launch. Non-traditional schools (e.g., vocational schools, private schools) composed a minority of responses and were removed for analysis to create a homogenous sample. Duplicate responses (i.e., those for whom identical districts were stated) were also omitted, with the most complete record being retained.

### 2.3. Statistical Analysis

Quantitative survey items were analyzed using the Statistical Package for Social Sciences (IBM SPSS Statistics for Windows, Version 27.0., Armonk, NY, USA: IBM Corp., 2020). Means, standard deviations, and frequencies (presented as valid percentages) were computed, as applicable, to summarize responses to each question. For questions that prompted the respondent for numeric data, responses were assessed for validity and grouped into categories for analysis. 

### 2.4. Thematic Analysis 

Open-ended survey responses were analyzed qualitatively using a codebook developed according to a standardized process, similarly to that outlined by MacQueen et al. [31]. The purpose of this analysis was explanatory in nature; the qualitatively-derived themes helped to make meaning from and provide context not otherwise captured in quantitative responses. One member of the team (AB) reviewed responses for development of an initial coding scheme, including a hierarchy of ‘parent’ and ‘child’ codes, which was then shared with a second member of the team (JG) who applied the coding schema to a subset of comments and made clarifying modifications to the codebook accordingly upon consensus-building conversation with the lead author (AB). The agreed upon coding schema was then used by the lead author to independently code and analyze all responses in NVivo (Version 12, Melbourne, Australia: QSR International Pty, Ltd., 2020). Individual district representatives were the unit of analysis so that themes were identified based on the frequency of experiences and perceptions across representatives within the sample, unbiased by repetitive commentary present in individual representatives’ responses across different questions.

## 3. Results

Representatives from a total of *n* = 209 districts were included in the final analysis, while *n* = 190 provided data enabling clear identification of their school typology (Table 1). Across school poverty typologies, response rates ranged from 19% (Rural High Poverty) to 50% (Urban Very High Poverty).

### 3.1. Quanitative Findings

Prior to COVID-19-related shutdowns, responding districts reported participating in numerous foodservice programs, including the National School Lunch Program (96.2%, *n* = 201), the School Breakfast Program (83.7%, *n* = 175) and the USDA/DOD Fresh Fruit and Vegetable Program (FFVP) (35.4%, *n* = 74). Overall, responding districts collectively served 3,371,489 breakfasts and 6,615,739 lunches in 2020. After school closures in March of 2020 and the typical Summer break, Fall classes resumed with districts largely adopting either hybrid (*n* = 153, 73.2%) or exclusively in-person (*n* = 50, 23.9%) education. Participation in child nutrition programs shifted; most schools leveraged the Seamless Summer Option (80.4%, *n* = 168) into the school year, with only a quarter of districts participating in the National School Lunch Program (24.9%, *n* = 52). 

Across seasons, many districts provided food or meals to local households in the community (Table 2). Specifically, the overwhelming majority (87.1%, *n* = 182) of responding districts reported providing food after the COVID-19-related shutdowns began in Spring, which decreased to 42.1% (*n* = 88) in the Summer. This continued into the Fall, with 15.3% of all districts (*n* = 32) providing food to the community via “mobile” meal services. Moreover, 18% (*n* = 28) of those that offered hybrid education during Fall provided “mobile” meal services, versus 8% (*n* = 4) of those offering an exclusively in-person education. 

When evaluating meal provision frequency and quantity of foods provided with more granularity, differences existed across seasons (Table 3). In Spring, it was most common for districts that continued foodservice operations to make meals available once weekly (43.4%, *n* = 79), with most (86.1%, *n* = 68) providing five days-worth of food at one time, per recipient. This approach remained common in Summer and Fall (45.5% (*n* = 40) and 53.1% (*n* = 17), respectively). Some within-season variability in meal service approach was noted (*n* = 17, *n* = 2, and *n* = 3 in Spring, Summer, and Fall, respectively), with responding districts providing meals at differing quantities and frequencies over the course of a given season. Most respondents’ descriptions of reasons for variability included mid-season modifications to align with changes in mode of education delivery, or decreasing frequency but increasing quantity (e.g., five meals provided once weekly versus one meal provided five times weekly) to limit staff exposure to COVID-19.

The majority of districts (67%, *n* = 122) that provided food in Spring functioned as “open sites,” distributing food to anyone from the community at large, and primarily via household pickup (i.e., households picked up food directly from the school or district office; 76.4%, *n* = 139). Though the number of schools providing meals to the community decreased in the Summer, the percentage of those functioning as open sites increased (87.5%, *n* = 77), while household pickup remained the most common method of distribution (76.1%, *n* = 67). In the Fall, among districts utilizing mobile meal delivery, most functioned as “restricted” (56.3%, *n* = 18) sites, meaning food was provided to only students in the district; 40.6% (*n* = 13) still operated as open sites. Household pickup of food remained the most common approach utilized in the Fall (84.4%, *n* = 27).

Nearly half (49.5%, *n* = 90) of districts that provided food to the community in Spring reported using a pre-order system to track demand for meals among those eligible. This system was most commonly used among those delivering food directly to households (64.7% vs. 47.5% among those employing meal pick-up). In Summer, this distribution remained similar, with 50% of districts offering meal delivery utilizing a preorder system compared to 40.3% of those offering meal pick up. The majority of districts offering mobile meals in Fall also employed pre-order systems (65.6%, *n* = 21), with meal delivery again remaining the most common approach for utilization of a pre-order system (72.7%, *n* = 8).

Across semesters, the majority of districts reported that 100% of their constituent schools applied for and received USDA meal pattern waivers (66.9%, 63.4%, and 70.8% of districts in Spring, Summer, and Fall, respectively). Qualitatively, districts that reported any portion of their schools applying for and using meal pattern waivers typically did so because of supply chain disruptions and difficulty obtaining necessary products. Additionally, in Fall 2020, the majority (75.8%, *n* = 122) of districts reported that 0% of their schools applied for and received USDA fresh fruit and vegetable program parent pickup waivers, which were first introduced in August of 2020 (i.e., the start of Fall 2020).

### 3.2. Qualitative Findings

Across optional open-ended prompts allowing for elaboration of close-ended questions throughout the survey, *n* = 158 (76%) respondents opted to provide further detail. For the open-ended, end-of-survey questions, *n* = 179 (86%) respondents provided insight on the primary factors that influenced their approach to COVID-related foodservice changes; *n* = 190 (91%) respondents detailed both what went well and what was challenging about their foodservice adaptations; and *n* = 182 (87%) respondents shared foodservice-related concerns for the coming school year. All open-ended responses were considered together for the thematic analysis. Analysis of the open-ended question responses resulted in the identification of four overarching themes, listed in order of the frequency with which they emerged across comments: (1) Challenges and Concerns, (2) Successes, (3) Service Adaptations, and (4) Reach (Table 4). 

Challenges and concerns were expressed at a frequency at least twice that of other themes, with supply chain issues, logistical difficulties, financial concerns, staffing issues, and COVID-related concerns (e.g., safety) emerging nearly equally as prominent sub-themes. Other challenges that emerged, albeit somewhat less frequently, included poor participation by families, and concerns about child and family well-being. 

General successes (‘Successes’) and approaches to adapting foodservice operations (‘Service Adaptations’) emerged with equal frequency, albeit about half as often as noted challenges. Prominent successes were staff adaptability and resilience, and mission achievement (i.e., in getting food to children and families who needed it). Less prominent across comments, but still important, were respondent’s mention of adhering to safety protocols and efforts to mitigate COVID exposure for foodservice staff and those they were serving. 

Service Adaptations captured sentiments related to the resources that districts leveraged to maintain their operations and the myriad factors that influenced why, when, and how they approached such adaptations. Districts were responsive to community and individual household needs (e.g., offering targeted delivery to households upon request); to the supply chain (e.g., changing meal or packaging plans with little to no notice); and, upon the start of a new school year in Fall 2020, to the learning modality (or modalities) of the schools in the district (i.e., remote, in-person, or hybrid and myriad combinations and fluctuations therein). How districts approached this operational feat varied widely both within and between districts and over time, as demonstrated by their close-ended question responses. Open-ended comments revealed the equal contribution of existing resources (e.g., communication channels, bus routes) and regulatory flexibilities (i.e., USDA waivers) to enabling such adaptations. 

With regard to Reach, districts reported relying on or collaborating with other entities (e.g., Salvation Army) to provide food to students and the community; targeting their efforts to ensure that high-risk subgroups were reached; and, expanding their scope to reach family members and households not typically served by school meals. 

The identified subthemes intersected in notable ways. For example, districts that worked to ensure high-risk subgroups of children were reached faced extra logistical hurdles and costs associated with navigating multiple modes of meal access (e.g., both pick-up sites and delivery routes). And while staff adaptability and mission achievement were both cited widely as key successes during this period, the more frequent and urgent discussion of foodservice program (in)viability in the context of staffing shortages and financial concerns suggests a degree of operational precarity predating the pandemic.

## 4. Discussion

### 4.1. Contextualizing the Findings

Throughout the COVID-19-related closures, schools in Ohio continued to provide meals despite substantial and unforeseen challenges. Many districts perceived this as an indisputable obligation given the heightened needs of the community; in fact, many districts opted to expand the scope of their foodservice operations to feed individuals throughout the community rather than focus solely on district-affiliated students as would normally be the case. These findings are largely consistent with nationally-representative data presented by the School Nutrition Association at the onset of the pandemic-related closures, which indicated among a geographically diverse sample of schools nationwide, the majority (95%) were providing “emergency meal assistance” despite staffing, financial, and supply chain challenges, among others [27]. However, the results were slightly inconsistent in that most (80.1%) were serving fewer meals than usual [27]. Given existing rates of food insecurity were exacerbated among key subgroups during COVID-19, including households with children, interruptions in this safety net could have critical effects on those already at risk, worsening food access, overall diet quality, and overall health [9,32]. 

While committed to serving their respective communities, responding district representatives voiced numerous frustrations regarding the COVID-19-related challenges they faced in doing so—in terms of safety, accessibility of needed supplies (e.g., PPE), and overall fears of the unknown that have coincided with the pandemic [30]. Other concerns represent shortcomings of both the larger systems on which districts are reliant—namely the food supply chain—and the site-level context of school foodservice operations (e.g., inability to store items due to inadequate space and equipment). Waivers provided by the USDA (e.g., meal pattern waivers) enabled flexibilities that helped districts to overcome the stated challenges, as well as others related to COVID-19 safety restrictions (e.g., reliance on shelf stable or pre-packaged items that could be packed in lunches easily and delivered in bulk to families to minimize exposure). There have been calls to maintain meal pattern requirements during closures in an effort to minimize disruptions to student diet quality [33]. However, these data suggest that waivers were instrumental in enabling districts to continue to provide meals in the context of substantial challenges. 

In line with the findings of Kenney et al., this study found that districts were flexible and resilient in their efforts to continue school foodservice operations. District approaches to continuing such operations—namely, the mode, frequency, and amounts of food provided—varied within- and across seasons, reflecting rapid response to the dynamic nature of the closures despite seemingly no preparation [34]. Adaptations were made to address the varying challenges faced, including those related to implementation [28], safety [30], and efficiency [29]. Such adaptations were achieved by utilizing existing resources in innovative ways, such as leveraging busses and their typical routes for meal delivery. In many instances, districts would not have been able to distribute food without such resources, particularly in rural districts where the response rate for this study was lowest (suggesting insufficient time and resources for ancillary tasks) and where districts faced the extra challenge of geographically-dispersed households. Other literature supports these findings on district resiliency and innovation—particularly related to where, how, and to whom meals are offered—and inspires consideration of what we can learn from such short-term innovations toward longer-term preparation for potential future disruptions [29,35]. 

The challenges faced by districts were exacerbated by the poor financial standing of their foodservice operations, with many foodservice departments fighting to remain viable. Getting families to fill out school meal applications, for example, was an existing challenge made more difficult during the pandemic and has ramifications for districts’ bottom lines [36]. Similarly, family and parent participation were commonly cited barriers, though in some instances this coincided with concerns that parents had no transportation, were unable to pick up meals because of work schedules, or had general concerns regarding COVID-19 exposure. Streamlining channels of communication and establishing external relationships with community partners could help boost participation, which several respondents in this study reported doing successfully [35]. 

The resilience and unrelenting concern for the students and families displayed by the foodservice workers was a noteworthy finding of this research. Similar work has pointed out that despite their essentiality, foodservice workers, as well as many others, are stigmatized and undervalued [37]. Across respondents, it was clear that foodservice workers were confronted with unpredictable and uncertain circumstances; often, such workers were overworked due to increased reach and logistical obstacles, while also facing fear of unemployment if schools went virtual, or if departments were unable to support employees financially. Such precarity contributed to challenges in maintaining staff morale, similarly noted elsewhere, though many voiced overall optimism particularly when able to cite their district’s success in feeding children [30]. This tension reflects broader societal conversations regarding undervalued and overworked frontline workers—a conversation that should be extended to include school foodservice staff.

### 4.2. Implications 

This work has important implications for research, practice, and policy. Since the beginning of the COVID-19-related shutdowns, several pieces of legislation have been introduced or enacted, and this study supports the need for and promise of these initiatives, particularly in states with high rates of food insecurity, such as Ohio. For instance, our findings demonstrate that waivers provided by the USDA were critical in enabling schools to continue providing meals; this includes meal pattern waivers and others (e.g., allowances for non-congregate feeding). While concerns do exist regarding effects of these measures on diet quality, additional action should be taken to address this issue more comprehensively. For example, efforts are needed to rectify the unstable food supply, particularly via strengthening of the local food system [38]. The Universal School Meals Program Act of 2021—in addition to advocating for free breakfast, lunch, dinner, and a snack to all school children—calls for incentivizing schools to utilize local food sources. Specifically, this legislation would result in schools receiving a $0.30 per meal incentive if they procure at least 25% of their food from local sources. The potential of this legislation is two-fold: forging relationships with local suppliers may help districts to mitigate the challenges of a global supply chain prone to instability, especially in times of (inter)national crisis, and existing literature indicates that universal free school meals support improved diet quality [3].

Executing local food initiatives, or helping schools to expand and sustain their reach in periods of emergency, requires close relationships with community partners [38]. This includes establishing partnerships with food banks, churches, mobile meal delivery programs, and others during periods of stability [34,39]. Such relationships may be particularly important in rural, geographically-dispersed districts characterized by high poverty rates. 

Additionally, changes are needed to district and school infrastructure to improve logistics not only for the purposes of emergency preparedness, but to streamline operations to ensure consistent financial stability. As noted elsewhere, these include innovative methods of communication between districts and families, but also between federal entities (i.e., USDA) and school districts. Further, supporting schools in obtaining and maintaining necessary equipment to store and prepare food, particularly perishable items conducive to improved diet quality, is essential. For example, the 2021 Build Back Better Act includes $500 million in school kitchen equipment grants in addition to expansion of the Community Eligibility Provision, while the School Food Modernization Act of 2021 similarly proposes financial assistance for obtaining equipment necessary for storage, preparation, and serving of food [40,41]. Similarly, lawmakers should take into consideration long-standing recommendations to help districts reduce school meal debt, including direct certification, reducing options for competitive foods, and others [36,42,43].

Efforts to improve other, broader mechanisms for supporting low-income families may also lead to improved access, diet quality, and resilience during periods of emergency. In 2021, the United States Department of Agriculture revised the Thrifty Food Plan, which sets the maximum SNAP benefit allotment, modernizing the plan in order to encompass “the cost of a healthy, practical diet on a limited budget” [44]. Now reevaluated every five years, the new plan is intended to reflect prices of a variety of healthy food items commonly consumed in the U.S. (e.g., canned food) [45]. Other approaches, such as expansion of P-EBT benefits, have also been advocated as approaches to encourage ease of access to high quality food during periods of emergency, such as that experienced during COVID-19 [15,33].

### 4.3. Limitations and Future Directions

This analysis is not without limitations. First, we focused on foodservice operations within public school districts in a single state: Ohio. Thus, our characterizations of school foodservice adaptations may not reflect the COVID-instigated foodservice adaptions of non-public school systems with different resource profiles nor of schools in other geographic regions, both within and outside of the United States. Furthermore, our anonymous response procedures did not afford the opportunity to evaluate how student-, staff-, and school-level variables may have influenced district decisions regarding COVID-related foodservice adaptations. Finally, we relied on survey responses from a single time point to characterize school district adaptations over multiple seasons: spring, summer, and fall 2020. This was done to reduce repeated response burden for participants and measures were taken during the survey design process to maximize clarity of wording and the validity of responses (e.g., organizing questions by season to reduce cognitive burden). Further, the incorporation of open-ended questions throughout and at the end of the survey afforded respondents the opportunity to explain nuances in their approach and to elaborate on their foodservice adaptation experiences in a manner that complemented the quantitative findings. 

Further research will be necessary to explore specific strategies that school districts can employ to enhance their foodservice resiliency for future emergencies and to understand the full spectrum of impacts experienced by students, schools, districts, and communities resulting from the documented foodservice adaptations. As appreciation for systems science and whole-of-community interventions grow, researchers have an opportunity to study and understand better how school-focused policies and programs may work in tandem with more broadly applicable policies to support community-wide food security and nutrition and health equity. 

## 5. Conclusions

During the COVID-19-related school closures, school districts experienced a call-to-action to feed students and the community at large. They faced substantial difficulty in doing so, but displayed notable resilience in their response and adaptation to these challenges. Their adaptations demonstrated innovation in foodservice (e.g., creating systems less reliant on equipment and leveraging existing resources) while highlighting timely opportunities to address long-standing issues in school foodservice operations for the sake of program viability, student diet quality, and the local food economy.

## Figures and Tables

**Table 1 ijerph-19-05991-t001:** Number of Districts Responding to COVID-19 Foodservice Survey by School Typology.

Typology Code ^a^	Full Descriptor	Count of Sponsor IRN (in Sampling Frame)	Count of Sponsor IRN In Survey Responses	Response Rate
1	Rural High Poverty	124	24	19%
2	Rural Average Poverty	107	38	36%
3	Small Town Low Poverty	111	34	31%
4	Small Town High Poverty	89	31	35%
5	Suburban Low Poverty	77	27	35%
6	Suburban Very Low Poverty	46	16	35%
7	Urban High Poverty	47	16	34%
8	Urban Very High Poverty	8	4	50%
		609	190	31%

^a^ Typology based on Ohio Department of Education School District Typology Definitions.

**Table 2 ijerph-19-05991-t002:** Food/Meal Provision and Methods of Distribution across Seasons.

	Spring % ^a^ (*n*) (*n* = 209)	Summer % ^a^ (*n*) (*n* = 209)	Fall % ^a^ (*n*) (*n* = 209)
**Service of Food or Meals to the Community**			
Yes	87.1 (182)	42.1 (88)	15.3 (32) ^b^
No	12.9 (27)	57.9 (121)	84.7 (177)
**Approach Taken for Food/Meal Provision ^c,d^**			
Open site—Anyone from the community could come for food	67.0 (122)	87.5 (77)	40.6 (13)
Restricted—Food was provided only to students in the district	31.9 (58)	15.9 (14)	56.3 (18)
Targeted—Food was prioritized for more ‘at-risk’ households in the district	6.6 (12)	3.4 (3)	3.1 (1)
Varied approach (i.e., ≥1 approach taken)	6.6 (12)	1.1 (1)	9.4 (3)
Other	1.1 (2)	3.4 (3)	3.1 (1)
Don’t Know	0.5 (1)	0.0 (0)	3.1 (1)
**Use of a Pre-Order System ^d^**			
Yes	49.5 (90)	37.5 (33)	65.6 (21)
No	50.5 (92)	61.4 (54)	34.4 (11)
Refuse to answer	0.0 (0)	1.1 (1)	0.0 (0)
**How Households Received Food/Meals ^c,d^**			
Delivered directly to all interested households in the district	37.4 (68)	31.8 (28)	34.4 (11)
Delivered to drop-off/pick-up point(s) in the community	43.4 (79)	39.8 (35)	40.6 (13)
Households picked them up at their school or district office	76.4 (139)	76.1 (67)	84.4 (27)
Varied approach (i.e., ≥1 way households could receive meals)	46.2 (84)	12.5 (11)	46.9 (15)
Other	0.0 (0)	0.0 (0)	0.0 (0)

^a^ Valid percent. ^b^ School was in session as of Fall 2020, data represents districts providing food via “mobile meal services”. ^c^ Respondents given the option to select ≥1 answer. ^d^ Among those who served food or meals to households.

**Table 3 ijerph-19-05991-t003:** Frequency and Quantity of Meals Provided ^a^.

	Spring Semester %^a^ (*n*) (*n* = 182) ^b^	Summer Semester % (*n*) (*n* = 88) ^b^	Fall Semester % (*n*) (*n* = 32)
**Food Served Every Weekday (Mon–Fri)**	** *n* ** ** = 50**	** *n* ** ** = 25 ^c^**	** *n* ** ** = 13 ^c^**
≤1 Meal	56.0 (28)	56.0 (14)	76.9 (10)
2 Meals	14.0 (7)	16.0 (4)	7.7 (1)
3 Meals	12.0 (6)	4.0 (1)	-
5 Meals	16.0 (8)	16.0 (4)	7.7 (1)
≥7 Meals	2.0 (1)	4.0 (1)	-
**Food Served 4 Times Per Week**	** *n* ** ** = 2**	** *n* ** ** = 2**	** *n* ** ** = 0**
≤1 Meal	50.0 (1)	50.0 (1)	-
3 Meals	50.0 (1)	-	-
Don’t Know	-	50.0 (1)	-
**Food Served 3 Times Per Week**	** *n* ** ** = 22**	** *n* ** ** = 6**	** *n* ** ** = 0**
≤1 Meal	4.5 (1)	-	-
2 Meals	63.6 (14)	66.7 (4)	-
3 Meals	18.2 (4)	33.3 (2)	-
5 Meals	13.6 (3)	-	-
**Food Served 2 Times Per Week**	** *n* ** ** = 50**	** *n* ** ** = 19**	** *n* ** ** = 5**
≤1 Meal	4.0 (2)	-	20.0 (1)
2 Meals	18.0 (9)	31.6 (6)	-
3 Meals	48.0 (24)	36.8 (7)	40.0 (2)
4 Meals	2.0 (1)	5.3 (1)	-
5 Meals	24.0 (12)	15.8 (3)	20.0 (1)
6 Meals	4.0 (2)	5.3 (1)	-
≥7 Meals	-	5.3 (1)	20.0 (1)
**Food Served Once Weekly**	** *n* ** ** = 79**	** *n* ** ** = 40**	** *n* ** ** = 17 ^c^**
≤1 Meal	1.3 (1)	-	5.9 (1)
3 Meals	2.5 (2)	-	-
5 Meals	86.1 (68)	72.5 (29)	76.5 (13)
≥7 Meals	8.9 (7)	25.0 (10)	11.8 (2)
Don’t Know	1.3 (1)	2.5 (1)	5.9 (1)

^a^ Frequency response options included: Food Served Every Weekday (Mon–Fri), 4 Times per Week, 3 Times per Week, 2 Times per Week, Once Weekly, and Less Than Once Weekly. Meal quantity response options included: ≤1 meal, 2 Meals, 3 Meals, 4 Meals, 5 Meals, 6 Meals, or ≥7 Meals. Rows not applicable to any schools in the sample have been omitted for concision. ^b^ Among those who reported serving food or meals to households. ^c^ One respondent provided no data as to frequency or refused to answer.

**Table 4 ijerph-19-05991-t004:** Summary of Qualitative Data Themes and Subthemes.

Subthemes and Definitions	Exemplar Quotes
**Theme 1: Challenges and Concerns**	
*Supply Chain*: Refers to reported challenges related to sourcing food and non-food items.	“We offered grab and go style only in case some households did not have the equipment on hand to reheat items…the ease of to go type meals that did not require cooking was needed for these households. [We did not have] a variety of menu items for our To Go Meals. We stick to the same menu weekly, which meets all meals pattern components. We also limited our ‘in person’ lunch menu to a two week rotation. We made both of these decisions to help with inventory and to help hedge against out of stock food items at our vendors. Sourcing food and staffing [have been challenging]. For example we may order prepackaged items….tomorrow all of the produce I ordered…will come in bulk. I called to confirm what I was getting on Tuesday, so I could prepare my staff. So tomorrow and Friday will be on a crunch to prepackage grape tomatoes, broccoli, and celery. We will also need to bag milk and juice as our meal pick up starts Monday at 8 am. We are in need of a new walk in cooler and walk in freezer at one of our schools. Our balances keeps getting lower and lower...many of our commodity items are our of stock. For example, frozen fruit cups, raisins, canned fruit, etc. the items usually cost us $2.95 a case and when we have to switch to a non commodity item we end up paying $30–$45 a case.”
*Logistical Difficulties:* Refers to logistical challenges related to meal preparation, service, and delivery.	“Logistics of having to package meals and distribute them in a nontraditional manner has been the greatest challenge. We also initially struggled with availability of packaging to use from approved vendors.”
*Finances and Assets*: Refers to financial difficulty and concerns regarding long-term program viability.	“Most challenging thing is getting people to fill out a free and reduced form because everyone is receiving free meals. This hurts our funding for next year.”
*Staffing and Manpower*: Refers to inadequate staffing or program challenges related to difficulties faced by staff.	“The biggest factor has been staffing issues….we only have one person to serve…”
*COVID Factors*: Refers to challenges related directly to COVID-19-related protocols.	“While the USDA reimbursement may be covering food expenses, there are many other factors now to successfully serving in COVID that are not covered. It has been challenging repeating to families that we want contactless delivery (please have your trunk available to us!) and wear a mask…”
*Family Participation*: Refers to challenges related to poor participation of parents, families, or children.	“Trying to get parents to take the free breakfast and lunches offered…”
*Children and Families*: Refers to general concern about the well-being and health of families and children.	“I think for the students it is best they be in school….with no school there is depression and other mental health issues along with there are more opportunities for the student to be out and about and spread the virus.”
**Theme 2: Successes**	
*Team Adaptability and Resilience*: Satisfaction with teamwork, collaboration, and staff resilience.	“Our team has definitely pulled together and adapted to a new non-traditional way of providing students access to our program.”
*Mission Achievement*: Refers to success in or general regard for serving children for the sake of doing it and being helpful.	“We distributed 42,000 kids in the spring with no COVID-19 exposure. The call to action to make sure no child was hungry was quick and seamless with my staff and administration.” “People in our district show great appreciation for the efforts we are making to feed kids”
*Safety Precautions*: Refers to successfully limiting exposure to COVID or success in following COVID precautions.	“Being very involved with our local health department, they have instructed us from the beginning on stricter serving techniques. The students do not touch items in the serving line. Therefore, our serving procedures have changed completely. Our foodservice staff ask every student their choices for lunch. This amount of time limits what our choices are….students are in assigned seats, and tables are released to come to the serving line to adhere to distancing. Sanitizing/disinfecting are critical. All tables and chairs are cleaning in between each of the 5 serving times.”
**Theme 3: Service Adaptations**	
*Adapting to Meet Needs*: Refers to adaptation or design of approach based upon reported needs.	“Meals were picked up unless a family did not have the means to do so. In that case we delivered to these families. We have made food available to all students in many different ways.”
*Adapting to Supply Chain*: Refers to necessary changes in approach due to issues with the food supply and supply chain.	“When other schools started serving only cold food, and we couldnt get bread, I started using food that was already in the freezer and we started serving hot food twice weekly on the pick up days and then provided two cold meals (or one cold meal depending on the day) to go with their hot food. We had a huge district response to hot food, our meals served skyrocketed and therefore we continued”
*Adapting to Learning Mode*: Refers to adaptation of delivery procedures in response to a change in mode of education delivery.	“This fall, we began delivering meals to our students when delivering the weekly educational packets to those students who were participating in remote learning.”
*Leveraging Existing Resources*: Refers to using existing tools, services, or technologies to continue foodservice operations.	“Families signed up on a google form and google form asked if you wanted delivery or pick up.”
*Leveraging USDA Waivers:* Refers to use of various waivers as a means to adapt approach in the face of challenges.	“Being able to be creative with our plans because the USDA has provided waivers and all kids eat free.”
**Theme 4: Strategic Reach**	
*Reliance or collaboration*: Refers to districts’ inability to help families and/or children, relying on others or directing them elsewhere.	“We tried to service as many families as we could…we had families pick up but their were some households unable to do pick up so we reached out to our local police and sheriff department who delivered meals to those in need…we helped families find other options for summer feeds that were available in July and August and I partnered with the [local food bank] to do once a week meal distribution for families.”
*Targeting of Efforts*: Refers to efforts specifically made to reach and/or target individuals who may be considered high-risk.	“We operated open sites but we also delivered to our specials needs children or to those folks that didn’t have transportation to the sites”
*Expansion of Scope*: Refers to efforts made by schools to reach individuals other than children.	“We opened meals to family members of the students in the district.”

## Data Availability

The data used in this study is available upon reasonable request.

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
