# Peer review of "Maintaining School Foodservice Operations in Ohio during COVID-19: “This [Was] Not the Time to Sit Back and Watch”"

_ijerph, 2022, doi:10.3390/ijerph19105991_

Round 1

Reviewer 1 Report

The article describes the ordeal of distributing food to the school children and families during the COVID lock down in the US state of Ohio.

This is an interesting narrative, which can guide for any future adversities. The authors have done a good analysis of information related to social status and food distribution patterns in the maximum school districts of Ohio.

Some of the background statistical information used for analysis looks to be very old or outdated like typology  code for districts (Table 10 which is 2013 data. This data is more than 7 years old when compared to the study period.

Information pertaining to food distribution presented in lines 161 to 163, I assume mean is with reference to each district. But I am sure that the population size in each district will be different. Unless the reader know this information, no meaningful interpretation can be drawn.

Another data which might be interesting is the age range of the children. This can give readers information on the influence of children on food collection and also parents responsibility in getting food. For example, for some of the high school kids, food might not have been collected or may be vice versa. Sex of the student also might have a influence on this process. But I do not see this reflected in the analysis.

The authors could also have reviewed some of the daily local news during the period to see the actual ground truth on this distribution pattern. Though this would be unconventional referencing, but that would certainly give some interesting and thoughtful information of the status and impact of this program. Including random interview of some parents and children.

Author Response

Please see the attachment for our full response.

Reviewer 2 Report

Dear Authors; This is an interesting study on the trend of school food services during pandemic. Exactly, in the point  I-as a North American licensed statistician- was getting excited by the results, its statistical analysis plan (SAP) had an unexpected  fatigue making the work impotent to deliver the key message to the government health policy makers! It lacks outcome difference over time subsection to make work statistically meaningful for government health policy makers ! This needs a "serious statistical addition" to make the work publishable in the journal, in my humble opinion. Please make sure to secure enough time from Journal editorial staff to add this missing gem. Regards.

P.S.

[1] Writing:

1-1 Author affiliations: Add the country (U.S.)

1-2 Line 129, 144: Remove the long stuff in the  () and add the following citations to the reference section:

[citation 1] IBM Corp. Released 2020. IBM SPSS Statistics for Windows, Version 27.0. Armonk, NY: IBM Corp

[citation 2] NVivo qualitative data analysis software; QSR International Pty Ltd. Version 12, 2020.

1-3 Nested Subsectioning: To make paper consistent, in "2.Materials&Methods" add subsection numbers: "2.1. Study Design & Survey Development", etc.

1-4 Discussion: Make its reading more smooth by breaking it down to three subsections: 4.Discussion; 4.1. This work; 4.2. Limitations and Contributions; 4.3. Future work

[2] Statistical:

2-1 For the used survey in the paper, what are reliability and validity statistics ? what is Cronbach's alpha ? Report it in the subsection "2.1. Study Design & Survey Development".

2-2 [Existentially Important !]    Add a subsection to results section titled "3.3. Temporal Change". In this important section, report the change of outcome results over time using the following logistic regression model(Without this new section, the paper fails to send the key message to government health policy makers!)

Model:    logit (P(Y=1))=b0+b1*Season+b2*controlling covariate

Y=1  service provided; =0 serviced missing

Season=0 Spring; =1 Summer; =2 Fall

controlling covariate: a covariate in the study deemed by you as a confounder

2-3 Trend Plot: Also, make sure to plot for P(Y=1) with 95%CI the trend over time in above analysis

*Reference for Logistic Regression:

https://en.wikipedia.org/wiki/Logistic_regression

Author Response

Please see attached for our full response. Thank you!

Reviewer 3 Report

  1. Line 39 mentions that 19% of children in the United States are marginal food security, and for high-risk children, school meals are associated with improved attendance, academic performance, and diet quality. In the Ohio state studied, does covid-19 affect high-risk children the same as children in general?
  2. Line 151 describe n=196 provided data enabling clear identification of their school typology, but n=190 in Table 1, the numbers don't match.
  3. How to Illustrate Rural High Poverty's response rate (19%), which is lower than other types.
  4. Table 3 based on the analysis of Frequency and Quantity of Meals Delivered, it is suggested to briefly explain whether Covid-19 has any impact on the classification of Table 1.
  5. Line 232 For the analysis of the results of the open-ended questionnaire, it is suggested to supplement the description of the overall response.
  6. Line 301 mentions that waivers were instrumental in enabling districts to continue to provide in the context of substantial challenges. So can Nationwide Waivers be the best way or strategy to address Child Nutrition? (both past and future)
  7. Line 359 A strong relationship between the school and community partners can be helpful in responding to emergencies such as covid-19. But what strategy does it maintain, and how can it improve its community energy and resilience in the Rural High Poverty communities?

Author Response

(The authors gave the same response as above.)

Round 2

Reviewer 2 Report

Dear Authors; Most of my concerns were addressed satisfactorily. Regarding, the inferential results on the logistic model, I think your argument is relatively valid but it is still fair for the readers to have them  as an Appendix before the reference section with an introductory paragraph on it. Regards.
